# Effects of Antithrombotic Treatment on Bleeding Complications of EBUS-TBNA

**DOI:** 10.3390/medicina57020142

**Published:** 2021-02-05

**Authors:** Hyun-Il Gil, Ryoung-Eun Ko, Kyungjong Lee, Sang-Won Um, Hojoong Kim, Byeong-Ho Jeong

**Affiliations:** 1Division of Pulmonary and Critical Care Medicine, Department of Internal Medicine, Kangbuk Samsung Hospital, Sungkyunkwan University School of Medicine, Seoul 03181, Korea; higil2004@gmail.com; 2Division of Pulmonary and Critical Care Medicine, Department of Medicine, Samsung Medical Center, Sungkyunkwan University School of Medicine, Seoul 06351, Korea; koryoungeun@gmail.com (R.-E.K.); kj2011.lee@samsung.com (K.L.); sangwonum@skku.edu (S.-W.U.); hjk3425@skku.edu (H.K.); 3Department of Critical Care Medicine, Samsung Medical Center, Sungkyunkwan University School of Medicine, Seoul 06351, Korea

**Keywords:** endobronchial ultrasound-guided transbronchial needle aspiration (EBUS-TBNA), antithrombotic agent, antiplatelet drug, anticoagulant agent, bleeding, complication

## Abstract

*Background and Objectives:* The application of endobronchial ultrasound-guided transbronchial needle aspiration (EBUS-TBNA) has been markedly increased over the past decade. EBUS-TBNA is known to be a very safe and accurate procedure; however, the incidence of bleeding complications in patients who are taking antithrombotic agents (ATAs) is not well established. *Materials and Methods:* We conducted a retrospective analysis of a prospectively registered EBUS-TBNA cohort in a single tertiary hospital from May 2009 to December 2016. The patients were divided into two groups: an insufficient discontinuation group, defined as having a prescription for ATAs on the procedure day or only interrupting them for a short period of time, and a sufficient discontinuation group, defined as having prescription for ATAs during 30 days prior to the procedure and interrupting them for a sufficient period of time. *Results:* During the study period, a total of 4271 patients, after excluding 3773 patients who did not take ATAs at all, 498 patients were classified into the insufficient discontinuation group (*n* = 102) and the sufficient discontinuation group (*n* = 396). The baseline characteristics of patients and examined lesions between two groups were not significantly different, except insufficient discontinuation group had longer prothrombin times than the sufficient discontinuation group. In the insufficient discontinuation group, the most common reasons for prescriptions of ATAs were ischemic heart disease (48.0%) and cerebral vascular disease (28.4%), and half of the patients were taking two or more ATAs. Eventually, only one bleeding complication in the insufficient discontinuation group (1/102, 1.0%) and one event in the sufficient discontinuation group (1/396, 0.3%) occurred (*p* = 0.368). *Conclusions:* EBUS-TBNA is considered a safe procedure in terms of bleeding complications, even in patients with insufficient stopping of ATAs.

## 1. Introduction

Over the past decade, endobronchial ultrasound-guided transbronchial needle aspiration (EBUS-TBNA) has become a routinely and easily applicable modality in the diagnosis or staging of pulmonary or mediastinal/hilar diseases, such as cancer, sarcoidosis, or abnormal lymphadenopathy. It was found that EBUS-TBNA is less invasive and as accurate as mediastinoscopy for the diagnosis and staging of lung cancer [1,2], and eventually, EBUS-TBNA became the first choice (when available) in the diagnostic process for lung cancer [3]. In particular, in the diagnosis and staging of clinical N1-3 non-small cell lung cancer (NSCLC), EBUS-TBNA has shown a diagnostic capability superior to that of mediastinoscopy [4]. Based on these results, as the number of patients requiring EBUS-TBNA increased, patients with multiple comorbidities in need of the procedure also increased. In some cases, patients with cardiovascular or cerebrovascular diseases who are taking antithrombotic agents (ATAs) may need to undergo EBUS-TBNA.

In the case of flexible bronchoscopic procedures undergone by patients taking ATAs, bleeding risk and management of ATAs (when to withhold and restart or the need for bridging agents) before and after the procedures are well established [5,6,7]. For the flexible bronchoscopic procedures, such as endobronchial biopsy (EBB) or transbronchial lung biopsy (TBLB), the discontinuation of clopidogrel seven days prior to consideration of biopsy is recommended, and low-dose aspirin alone can be continued [7,8]. However, the management of ATAs before and after EBUS-TBNA has not yet been well understood. Several previous studies have shown that EBUS-TBNA could be performed safely in patients taking clopidogrel, but these studies were based on small numbers of cases [9,10,11].

Therefore, the aim of this study was to compare the incidence of bleeding complications of EBUS-TBNA according to antithrombotic therapy in a large number of cases.

## 2. Materials and Methods

### 2.1. Patients

We conducted a retrospective cohort study based on the prospectively collected EBUS-TBNA registry data of the Samsung Medical Center (a 1989-bed, university-affiliated, tertiary referral hospital in Seoul, South Korea) from May 2009 to December 2016. All the patients who underwent EBUS-TBNA were to be included. The cases that met the following criteria were excluded: (1) cases of EBB and/or TBLB performed simultaneously in the same session of EBUS-TBNA and (2) cases without information on the use of ATAs.

The Institutional Review Board of the Samsung Medical Center approved this study (IRB no. 2020-01-062, 3 February 2020) and waived the requirement for informed consent because of the observational nature of the research. Additionally, the patients’ information was anonymized and de-identified prior to the analysis.

### 2.2. Data Collection and Definitions

The following data were gathered using the database: patient-related factors such as age, sex, preprocedure diagnosis, laboratory values, and history of ATA use; procedure-related factors such as the examined site, the size of the examined lesion on the transverse plan of computed tomography (CT) images, the number of needle passes per lesion, and the number of obtained cores per lesion, and procedure time; and the details of bleeding complications.

Since the recommended period for how long ATAs should be discontinued prior to the EBUS procedure in patients receiving ATAs is not clear, we defined the “insufficient discontinuation group” for ATAs when each drug was continuously used or discontinued for less than the following periods according to the criteria of the recommended guidelines in the field of flexible bronchoscopy [8]: (1) if clopidogrel (an irreversible P2Y_12_ receptor blocker; antiplatelet drug) was discontinued for less than 5 days; (2) if cilostazol (phosphodiesterase-3 inhibitor; reversible antiplatelet drug) was discontinued for less than 2 days; (3) if warfarin (inhibits production of vitamin K-dependent clotting; oral anticoagulant) was discontinued for less than 5 days; (4) if a low-molecular-weight heparin (LMWH; antithrombin inhibitor), such as enoxaparin and dalteparin, was discontinued for less than 24 hr; or (5) if direct oral anticoagulants (DOACs; factor Xa inhibitors), such as rivaroxaban, edoxaban, and apixaban were discontinued for less than 2 days. Due to the fact that aspirin (irreversible cyclooxygenase inhibitor; antiplatelet drug) is not recommended to be stopped before the EBB and/or TBLB, even if aspirin was continuously used before the procedure, the case was not classified as belonging to the insufficient discontinuation group. There was no history of using ATAs other than those listed above in this EBUS cohort.

Patients who did not belong to the insufficient discontinuation group were divided into the no ATAs and sufficient discontinuation groups. The no ATAs group was defined as no history of using ATAs within 1 month prior to the EBUS procedure, and the sufficient discontinuation group was defined as patients who used ATAs within 1 month but had discontinued them for a sufficient period as mentioned above.

We defined the bleeding complications as the need to interrupt the EBUS procedure and the requirement for endobronchial epinephrine instillation, red blood cell transfusion, or an increased level of care after EBUS-TBNA. Mild bleeding complications, such as those easily controlled by cold saline instillation or spontaneous hemostasis, were not considered significant bleeding complications.

### 2.3. EBUS-TBNA Procedures

Details of the EBUS-TBNA procedure were described in our previous reports [12,13]. Briefly, EBUS-TBNA was performed using a convex probe-EBUS bronchoscope and a 22-gauge needle. Patients were moderately sedated using intravenous midazolam with or without fentanyl with intratracheal topical 2% lidocaine. Three passes per lesion were attempted, and there were at least two passes when the core tissue was obtained.

### 2.4. Statistical Analysis

Data are presented as the means ± standard deviations or the medians and interquartile ranges for continuous variables, and as numbers and percentages for categorical variables. Continuous variables were compared using the Mann–Whitney *U* test, and categorical variables were compared using the Pearson χ^2^ test or Fisher’s exact test. All analyses were performed using R Statistical Software (Version 3.2.5; R Foundation for Statistical Computing, Vienna, Austria). All tests were two-sided, and a *p*-value less than 0.05 was considered statistically significant.

## 3. Results

During the study period, a total of 4859 cases underwent EBUS-TBNA. Of these, 579 (11.9%) cases simultaneously received EBB and/or TBLB, and nine (0.2%) cases without adequate information to identify the history of ATAs were excluded (Figure 1). A total of 4271 cases were identified, including 102 (2.4%) cases with an insufficient discontinuation group, 396 (9.3%) cases with a sufficient discontinuation group, and 3773 (88.3%) cases with a no ATAs group.

### 3.1. Basline Characteristics

The baseline characteristics of the two groups (insufficient discontinuation and sufficient discontinuation groups) at the time of the EBUS-TBNA procedures are shown in Table 1. The demographic characteristics of both groups were not statistically different. Preprocedure diagnoses were similar in both groups, and most of them underwent EBUS procedures for the differential diagnosis of lung cancer.

In the laboratory results, there were no statistical differences in the platelet counts, activated partial thromboplastin times (aPTTs), blood urea nitrogen, creatinine, and liver function tests between the two groups. However, the insufficient discontinuation group had longer prothrombin times (PT) (international normalized ratio [INR]: 1.1 vs. 1.0, *p* = 0.026) than that of the sufficient discontinuation group.

The EBUS procedure details are shown in Table 2. A total of 284 lesions were examined in the insufficient discontinuation group and 999 lesions were examined in the sufficient discontinuation group. There were no statistical differences in the sites examined, total procedure times, the sizes of the examined lesions, number of needle passes per lesion, or number of obtained core tissues per lesion between groups. Of all, the most commonly examined site was the mediastinal lymph nodes (82.1%). The mean size of the examined lesions, for all 1283 lesions, was 11.8 mm for the short axis and 16.8 mm for the long axis. In addition, the mean procedure time for all 498 patients was 21.0 min.

### 3.2. Antithrombotic Medication

The histories of ATA medication are shown in Table 3. In the insufficient discontinuation group, the most common reason for prescribing ATAs was ischemic heart disease (48.0%), followed by cerebral vascular disease (28.4%) and prevention (11.8%). The most common ATA administered was clopidogrel (84.3%), followed by aspirin (44.1%) and cilostazol (12.7%). In addition, half of the subjects (52/102, 51.0%) were prescribed two or more ATAs; the most commonly taken combination was aspirin plus clopidogrel (42/52, 80.8%).

In the sufficient discontinuation group, the most common reason for prescribing ATAs was prevention (35.6%), followed by ischemic heart disease (30.6%) and cerebral vascular disease (20.2%). The most common ATA prescribed was aspirin (83.1%), followed by clopidogrel (14.4%) and cilostazol (5.6%). Of 396 patients, 49 patients (12.4%) were prescribed two or more ATAs, and the most commonly taken combination was aspirin plus clopidogrel (33/49, 67.3%).

### 3.3. Bleeding Complications

Table 4 lists the cases of bleeding complications. Of all 4271 cases, there were only eight (0.2%) cases that presented bleeding complications. There was no statistical difference in the rates of bleeding complications between two groups (1/102 [1.0%] in the insufficient discontinuation group vs. 1/396 [0.3%] in the sufficient discontinuation group, *p* = 0.368).

In the insufficient discontinuation group, one patient (patient #1), who was prescribed aspirin plus clopidogrel for ischemic heart disease, presented bleeding complications and subsequently received a red blood cell transfusion. He complained of persistent hemoptysis and dyspnea after the procedure. He showed low oxygen saturation and oxygen was supplied via high-flow nasal cannula. Other vital signs were stable except desaturation. There were no specific abnormal findings in chest x-rays taken 4, 8, and 18 h after the procedure. On the day after the procedure, the hemoglobin decreased from 10.5 g/dL to 8.7 g/dL, and one pack of red blood cell was transfused. One the next day, hemoptysis and dyspnea improved, oxygen demand decreased, and the patient stabilized without any further treatment other than blood transfusion. In the sufficient discontinuation group, one patient (patient #2) showed bleeding after EBUS-TBNA, but it was controlled by endobronchial epinephrine instillation. He had a warfarin prescription for thrombotic arrhythmia, but stopped the medication nine days before the procedure, and the prothrombin time was within the normal range. The other six patients had no ATA medication history and no abnormal laboratory findings. However, of these seven patients in the sufficient discontinuation group or no ATAs group, four patients had large portions of necrosis in contrast CT images at the examined lesion of the EBUS-TBNA, and two patients had mucosal fragility due to suspected tumor infiltration in the gross bronchoscopic findings at the site of the EBUS-TBNA. Most bleeding complications were controlled by endobronchial epinephrine instillation (six of eight patients; 75%). One patient (patient #8) in the no ATAs group moved to intensive care unit (ICU) after EBUS-TBNA. She received EBUS-TBNA for right lower paratracheal node, subcarinal node, and right interlobar nodes. After the procedure, she showed desaturation, hypotension and high fever. Six hours later, her hemoglobin dropped from 10.3 g/dL to 8.2 g/dL and the chest x-ray taken 4 h after EBUS-TBNA showed diffuse haziness on right lung. Under the impression of post-EBUS bleeding and shock with pneumonitis, she moved to the ICU. The patient recovered quickly and was discharged three days later. None required intubation or a massive transfusion. Except for patient #8, there were no specific findings on the routine post-EBUS chest x-ray and CT was not performed in all patients. All of the patients survived until discharged.

The comparison between the insufficient discontinuation group and combining the sufficient discontinuation plus no ATAs groups is in the online Appendix A. There was no statistical difference in the rates of bleeding complications between groups (1/102 [1.0%] in the insufficient discontinuation group vs. 7/4169 [0.2%] in the sufficient discontinuation plus no ATAs groups, *p* = 0.176).

## 4. Discussion

To the best of our knowledge, this is the largest study focused on bleeding complications of EBUS-TBNA. Of 4271 patients, 498 (11.7%) patients were taking ATAs within one month of the EBUS procedure. Of these 498 patients, 102 (20.5%) patients were unable to discontinue the ATAs for a sufficient period of time because the diagnosis could not be delayed or the risk was too high to stop the ATAs. The insufficient discontinuation group seemed to have more hemorrhagic tendencies with prolonged INR, however, these differences were very small (1.1 vs. 1.0, not clinically significant). The main reasons for ATA medication were ischemic heart disease and cerebral vascular disease in the insufficient discontinuation group, and half of these patients were taking aspirin plus clopidogrel. Although there were difficulties in statistical verification due to the small incidence of the primary outcome, the bleeding complication rates were very low in both groups (1.0% in the insufficient discontinuation group and 0.3% in the sufficient discontinuation group).

Many patients who require histological diagnosis are often forced to delay the procedure because they are taking ATAs that can promote bleeding. As shown in our results, nearly 60% of patients were taking ATAs for secondary prevention purposes (to prevent another thrombotic event for people who have had thrombotic diseases), not just for primary preventive purposes (to prevent people from developing cardiovascular diseases). If an operator decides to discontinue ATAs for the EBUS procedure, it should be kept in mind that not only will the EBUS procedure be delayed, but the risk of certain vascular diseases is also increased by discontinuing the ATAs. In particular, the discontinuation of ATAs immediately after percutaneous coronary intervention significantly increases the risk of acute cardiac complications [14]. In the case of EBUS-TBNA in patients with a recent history of acute coronary syndrome, one of our studies analyzed the complications of EBUS-TBNA in the patients receiving ATAs within one year after percutaneous coronary intervention. Even though this study showed no significant bleeding complications after EBUS-TBNA in subjects on ATAs, the number of patients was small [15]. Not only cardiovascular complications were involved; there was a significant increase in both stroke and major vascular events within 30 days of medication discontinuation for any reasons, and the event rate peaked in the first three to nine days after withdrawal [16,17].

We analyzed patients who underwent EBUS-TBNA only and excluded those who underwent EBB and/or TBLB in the same session, because EBB and/or TBLB were identified as risk factors for complications of EBUS-TBNA in previous studies by Eapen et al. [18]. In our study, the frequency of significant bleeding complications after EBUS-TBNA was very low at 0.2% (8 of 4271 total cases). This is consistent with the results of a previous study, which also reported a bleeding complications rate of 0.2% (3 of 1317 EBUS-TBNA cases) [18]. It is noteworthy that bleeding complications were very low at 1.0% (1/102) even in the insufficient discontinuation group. Although there was a statistical difference in the baseline characteristics reflecting bleeding tendency (PT INR), the difference was very small and seemed to have no clinical meaning. Above all, since six out of eight patients with bleeding complications had bronchoscopic (mucosal fragility) or CT findings (large portions of necrosis) that could lead to bleeding at the EBUS-TBNA site, it would be more important to understand the characteristics of the EBUS-TBNA site to be tested than laboratory tests. In addition, since bleeding complications occurred while taking two ATAs in the insufficient discontinuation group, patients taking two or more ATAs should be managed with more caution or, if possible, it would be safer to take only one ATA and stop the others.

There are several limitations to our study. First, this was a retrospective study conducted in a single institution. Thus, this study has a selection bias and may have limitations in generalizability. Second, significant statistical analysis was limited because of the overly small number of complication events. However, the data were collected in a prospectively registered cohort, and we think the number of patients was large enough to compensate for these disadvantages. Third, in this study, ATAs were maintained or discontinued in consideration of the patient’s urgency and the risk of stopping ATAs according to each operator without specific guidelines. Therefore, as operators would have actively discontinued ATAs or simplified the procedure as much as possible for patients who were expected to have a high risk of bleeding, caution should be taken in interpreting the results of this study. Finally, for warfarin, as suggested in the previous guidelines [8], we determined whether patients should be included in the insufficient discontinuation group based on discontinuation five days before the procedure day. However, in real clinical practice, coagulation tests will be more important. Usually, patients with PT INR > 1.5 or aPTT > 50 s are judged to have a hemorrhagic tendency. If this were applied to the patients receiving warfarin in our study, the number of patients showing hemorrhagic tendencies on the day of the procedure was four of five (80.0%) in the insufficient discontinuation group and 3 of 22 (13.6%) in the sufficient discontinuation group. Even if the group were divided based on the coagulation test results, not the interruption period for warfarin, there would be almost no change from the previous results at 1/104 (1.0%) in the insufficient discontinuation group versus 1/398 (0.3%) in the control group (*p* = 0.372).

## 5. Conclusions

In conclusion, EBUS-TBNA is considered a safe procedure in terms of bleeding complications, even in patients taking ATAs or not stopping them for a sufficient period of time. Therefore, in patients risking a high possibility of thromboembolic events by discontinuing ATAs or those requiring a quick diagnosis, EBUS-TBNA without discontinuation of ATAs can be performed relatively safely.

## Figures and Tables

**Figure 1 medicina-57-00142-f001:**
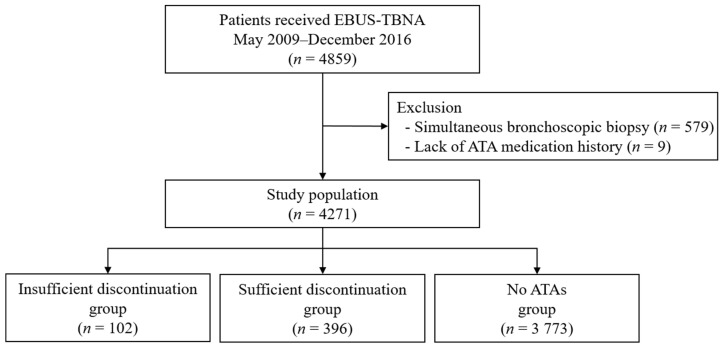
Flow sheet. EBUS-TBNA = endobronchial ultrasound-guided transbronchial needle aspiration; ATA = antithrombotic agent.

**Table 1 medicina-57-00142-t001:** Baseline characteristics of patients who received EBUS-TBNA.

Variables	Insufficient Discontinuation Group (*n* = 102)	Sufficient Discontinuation Group (*n* = 396)	*p*
Age, years	69.7 ± 7.2	68.6 ± 7.9	0.201
Sex, male	84 (82.4)	316 (79.8)	0.661
Pre-procedure diagnosis			0.526
Primary lung cancer	91 (89.2)	342 (86.4)	
Other cancer	8 (7.8)	37 (9.3)	
Other benign disease	3 (2.9)	17 (4.3)	
Laboratory tests			
Platelet count, ×10^3^/µL	252.6 ± 101.3	247.1 ± 81.3	0.924
Prothrombin time, INR	1.1 ± 0.3	1.0 ± 0.1	0.026
aPTT, seconds	37.7 ± 7.7	36.9 ± 6.5	0.459
Blood urea nitrogen, mg/dL	17.6 ± 7.2	17.0 ± 6.5	0.850
Creatinine, mg/dL	1.1 ± 0.7	1.0 ± 0.8	0.174
Total bilirubin, mg/dL	0.5 ± 0.3	0.5 ± 0.2	0.547
Aspartate aminotransferase, U/L	22.7 ± 12.2	21.6 ± 8.4	0.946
Alanine aminotransferase, U/L	20.2 ± 14.8	19.7 ± 11.7	0.932

Data are presented as number (percentage) or as mean ± standard deviation. EBUS-TBNA = endobronchial ultrasound-guided transbronchial needle aspiration; INR = international normalized ratio = aPTT, activated partial thromboplastin time.

**Table 2 medicina-57-00142-t002:** Characteristics of examined lesions.

Variables	Insufficient Discontinuation Group (*n* = 284)	Sufficient Discontinuation Group (*n* = 999)	*p*
Examined site			0.897
Mediastinal LN	231 (81.3)	822 (82.3)	
Hilar, interlobar, and lobar LN	42 (14.8)	137 (13.7)	
Others *	11 (3.9)	40 (4.0)	
Size of examined lesion, mm ^†^			
Short-axis diameter	11.2 ± 6.7	12.0 ± 6.9	0.093
Long-axis diameter	16.2 ± 9.6	17.0 ± 9.8	0.211
Number of needle passes per lesion	1.9 ± 0.8	1.9 ± 1.1	0.389
Number of obtained core tissues per lesion	1.5 ± 0.7	1.6 ± 1.1	0.128
Procedure time, minutes	21.7 ± 12.1	20.8 ± 11.7	0.640

Data are presented as number (percentage) or as mean ± standard deviation. LN = lymph node. * Lung parenchymal lesions and pleural seeding nodules were included. ^†^ Size on the transverse plane of computed tomography images.

**Table 3 medicina-57-00142-t003:** Antithrombotic medication.

Variables	Insufficient Discontinuation Group (*n* = 102)	Sufficient Discontinuation Group (*n* = 396)	*p*
Reason for antithrombotic medication			
Ischemic heart disease	49 (48.0)	121 (30.6)	0.001
Cerebral vascular disease	29 (28.4)	80 (20.2)	0.090
Prevention	12 (11.8)	141 (35.6)	<0.001
Thrombotic arrhythmia	7 (6.9)	39 (9.8)	0.474
Peripheral artery occlusive disease	4 (3.9)	6 (1.5)	0.125
Other *	1 (1.0)	9 (2.3)	0.695
Antithrombotic medication			
Aspirin	45 (44.1) †	329 (83.1)	<0.001
Discontinued days	1 [0–4]	6 [5–8]	-
Clopidogrel	86 (84.3)	57 (14.4)	<0.001
Discontinued days	1 [0–5]	7 [6–7]	-
Cilostazol	13 (12.7)	22 (5.6)	0.019
Discontinued days	1 [0–2]	6 [3–6]	-
Warfarin	5 (4.9)	22 (5.6)	0.807
Discontinued days	2 [1–4]	6 [6–7]	-
LMWH	4 (3.9)	7 (1.8)	0.246
Discontinued hours	19 [17–26] ‡	30 [27–37]	-
Direct oral anticoagulants	5 (4.9)	5 (1.3)	0.034
Discontinued days	1 [1–1]	5 [2–7]	-
More than two drugs	52 (51.0)	49 (12.4)	<0.001

Data are presented as number (percentage) or as median (interquartile range). LMWH = low molecular weight heparin. * Others include valvular heart disease (*n* = 4), pulmonary thromboembolism (*n* = 3), essential thrombocytosis (*n* = 2), and superior vena cava syndrome (*n* = 1). ^†^ All 45 of these patients were taking more than two antithrombotic agents, including aspirin. In other words, these patients were classified into the insufficient discontinuation group not just because they were taking aspirin, but because they were also taking another ATA. ^‡^ Of these four patients, three patients last used the LMWH 12 h, 18 h, and 19 h before the procedure, respectively. One patient last used the LMWH 47 h before the procedure, but was classified into the insufficient discontinuation group because clopidogrel was used up to two days before the procedure.

**Table 4 medicina-57-00142-t004:** Details of cases with bleeding complications.

Case #	Group	Age/ Sex	Reason for Procedure	Platelets, ×10^3^/µL	PT, INR	No. of Examined Lesions, *n*	Singularity on the Examined Lesions	Procedure Time, min	Management for Bleeding	Final Diagnosis
#1 *	Insuff	68/M	Suspected meta of CRC	363	1.18	3	-	28	TF	Meta of CRC
#2 ^†^	Suff	74/M	Suspected LC	296	1.00	1	Necrotic	20	Epi	NSCLC (cT4N1M0)
#3	No ATAs	68/F	Suspected LC	251	0.95	4	Necrotic	40	Epi	Benign granuloma
#4	No ATAs	66/M	Suspected LC	308	1.03	3	Mucosal fragility	14	Epi	NSCLC (cT2aN2M0)
#5	No ATAs	74/F	Suspected recurrence of LC	125	1.01	2	-	17	Epi	No recurrence
#6	No ATAs	49/M	Suspected LC	387	1.09	2	Mucosal fragility	19	Epi	NSCLC (pT3N1M0)
#7	No ATAs	66/F	Suspected LC	266	1.07	3	Necrotic	14	Epi	NSCLC (cT3N2M0)
#8	No ATAs	60/F	Suspected meta of EMC or HCC	163	1.04	3	Necrotic	10	ICU	Meta of EMC

PT = prothrombin time; INR = international normalized ratio; Insuff = insufficient discontinuation group; M = male; meta = metastasis; CRC = colorectal cancer; TF = transfusion of red blood cells; Suff = sufficient discontinuation group; LC = lung cancer; Epi = endobronchial instillation of epinephrine; NSCLC = non-small cell lung cancer; ATAs = antithrombotic agents; F = female; EMC = endometrial cancer; HCC = hepatocellular carcinoma; ICU = intensive care unit. * This patient was taking aspirin plus clopidogrel for ischemic heart disease, and received the procedure with only one day off from antithrombic agents. ^†^ This patient was taking warfarin for atrial fibrillation, and underwent the procedure with warfarin off for 9 days.

## Data Availability

Data and material are available on reasonable request.

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
