# Peer review of "Effects of Antithrombotic Treatment on Bleeding Complications of EBUS-TBNA"

_medicina, 2021, doi:10.3390/medicina57020142_

Round 1

Reviewer 1 Report

In the manuscript, the authors present a study regarding the impact of the anticoagulant treatment in patients in which EBUS-TBNA is performed. They included in the study 4271 subjects that were divided in 2 groups : : an insufficient discontinuation group (102 patients) and the control group (4169 patients). In my opinion it is an interesting manuscript that can be very useful in the clinical practice. In order to improve the quality of the manuscript, in my opinion some changes of the manuscript have to be done. My observation are :

1 In the manuscript, the authors stated that in 6 cases of bleeding complication (from 8 cases) the only medical management was the endobronchial epinephrine instillation. Please specify what happened with the other 2 cases. 

2. In the cases of bleeding complications do you performed thoracic CT scan ? Did you notice any cases of mediastinal haematoma or haemothorax ? 

Author Response

Reviewer #1

C1. In the manuscript, the authors present a study regarding the impact of the anticoagulant treatment in patients in which EBUS-TBNA is performed. They included in the study 4271 subjects that were divided in 2 groups : : an insufficient discontinuation group (102 patients) and the control group (4169 patients). In my opinion it is an interesting manuscript that can be very useful in the clinical practice. In order to improve the quality of the manuscript, in my opinion some changes of the manuscript have to be done. My observation are :

R1. Thank you for your thoughtful comments to improve the quality of this report.

C2. In the manuscript, the authors stated that in 6 cases of bleeding complication (from 8 cases) the only medical management was the endobronchial epinephrine instillation. Please specify what happened with the other 2 cases. 

R2. For the patient #1, he received EBUS-TBNA for 4R, 7, 4L nodes. After the procedure, he complained of persistent hemoptysis and dyspnea after the procedure. His SpO2 was decreased, and oxygen was supplied via high-flow nasal cannula. There were no specific abnormal findings in chest X-rays taken 4 hours, 8 hours, and 18 hours after the procedure. On the day after the procedure, the hemoglobin decreased from 10.5 g/dl to 8.7 g/dl, and one pack of RBC was transfused. On the next day, hemoptysis and dyspnea improved, O2 demand decreased, and the patient stabilized without any further treatment other than blood transfusion.

For the patient #8, she received EBUS-TBNA for 4R, 7 and 11Ri nodes. After the procedure, she complained persistent hemoptysis, and showed desaturation, hypotension and high fever. Six hours after the procedure, her hemoglobin was dropped from 10.3 g/dl to 8.2 g/dl. Post-EBUS chest x-ray also showed increased haziness at right lung field. Under the impression of post-EBUS bleeding and shock with pneumonia, she moved to the ICU and received EGDT (early goal directed therapy; including volume resuscitation, vasopressor and antibiotics). Vasopressor was used for a total of 1 day, and a length of stay in ICU was 3 days. This cases was occurred in 2013 when EGDT was standard of care against septic shock.

We described these cases in the revised manuscript (Line 202-211, 219-226).

C3. In the cases of bleeding complications do you performed thoracic CT scan ? Did you notice any cases of mediastinal haematoma or haemothorax ? 

R3. All 8 patients underwent chest x-ray within 4 hours after the procedure. However, CT was not performed because there was no significant change in the x-ray after the procedure except for patient #8 mentioned above in C2. The patient #8 moved to ICU for septic shock management, and improved quickly after ICU care, so CT scan was not performed.

We added this information in the revised manuscript (Line 227-229).

Reviewer 2 Report

Dr. Gil and colleagues from Korea performed a retrospective analysis on the bleeding complications of EBUS-TBNA. Patients were divided into two groups: the insufficient discontinuation group and the control group. The latter one consisted of two subgroups: sufficient discontinuation vs. no antithrombotic agent group. They found that the prevalence of bleeding complications was low in both the insufficient discontinuation and control groups.

It is a well-written study addressing an important clinical question. These data are useful in guiding the future clinical practice with regards to EBUS-TBNA.

My major comment is the authors should consider using the “sufficient discontinuation” group as the control group, rather than combining it with the no-ATA group. The fundamental question is we don’t know whether maintaining antithrombotic/discontinued insufficiently will increase the risk of major bleeding among these patients. If the authors can remove the patients previously not on ATAs and then compare the “insufficient discontinuation” group to the “sufficient discontinuation,” the comparison would be more clinically meaningful.

Also, if possible, the authors may consider using propensity-score matching or exact matching method to identify two (even more) comparable groups (matched “insufficient discontinuation” and “sufficient discontinuation” groups) and compare their outcomes. As a retrospective study, there is no way to fully prevent the selection bias, but matching may be a good way to mitigate the potential confounding.

Besides, as the authors have extensive experience in EBUS-TBNA, the authors may consider sharing their experience (in the discussion section) on identifying the high-risk patients who should cease ATA long enough to prevent the major bleeding.

Author Response

Reviewer #2

C1. Dr. Gil and colleagues from Korea performed a retrospective analysis on the bleeding complications of EBUS-TBNA. Patients were divided into two groups: the insufficient discontinuation group and the control group. The latter one consisted of two subgroups: sufficient discontinuation vs. no antithrombotic agent group. They found that the prevalence of bleeding complications was low in both the insufficient discontinuation and control groups.

It is a well-written study addressing an important clinical question. These data are useful in guiding the future clinical practice with regards to EBUS-TBNA.

R1. Thank you for your thoughtful comments to improve the quality of this report.

C2. My major comment is the authors should consider using the “sufficient discontinuation” group as the control group, rather than combining it with the no-ATA group. The fundamental question is we don’t know whether maintaining antithrombotic/discontinued insufficiently will increase the risk of major bleeding among these patients. If the authors can remove the patients previously not on ATAs and then compare the “insufficient discontinuation” group to the “sufficient discontinuation,” the comparison would be more clinically meaningful.

R2. Thanks for the great comment. We agree with your opinion. In Tables 1 and 2 comparing the characteristics of both groups, only the “insufficient discontinuation group” and the “sufficient discontinuation group” were compared again, excluding patients receiving no ATA medication. In this revised comparison, the occurrence of the bleeding complication events is observed as 1/102 (1.0%) and 1/396 (0.3%), and the p value still showed no statistical difference. We rewrote it in the revised manuscript (Whole through the manuscript including the Abstract, Table 1 and 2, Figure 1, and New Online supplement file).

C3. Also, if possible, the authors may consider using propensity-score matching or exact matching method to identify two (even more) comparable groups (matched “insufficient discontinuation” and “sufficient discontinuation” groups) and compare their outcomes. As a retrospective study, there is no way to fully prevent the selection bias, but matching may be a good way to mitigate the potential confounding.

R3. Thanks for your comment. Because of huge difference in numbers between the comparing groups, we also thought that propensity-score matching or exact matching methods would be necessary. We discussed for this problem with our biostatisticians. However, since there are very few outcome events, we concluded that matching methods did not mean much, and it is better to compare all patients than matching. (In general, when matching methods are applied, some cases that cannot be matched are bound to be lost in the final analysis model. So, as in this paper, if there are few outcome events, applying matching methods may result in loss of outcome events, which may cause more difficulty in analysis.) And, we have already discussed in the Discussion section about the limitation that it is difficult to perform statistical analysis because there are few outcome events.

C4. Besides, as the authors have extensive experience in EBUS-TBNA, the authors may consider sharing their experience (in the discussion section) on identifying the high-risk patients who should cease ATA long enough to prevent the major bleeding.

R4. Although the incidence of bleeding complication was very small, the analysis of 8 patients with bleeding complications showed that bleeding occurred when mucosal fragility (gross findings on bronchoscopy) was present at the puncture site or target lesion was necrotic (on chest CT scan). Before EBUS-TBNA, operator should check the CT and gross bronchoscopic finding, and bleeding complication should be noted when such findings are present. In addition, since one bleeding case in the “insufficient discontinuation group” was taking dual antiplatelet agents, patients taking two or more drugs should be managed with more caution or, if possible, discontinue one drug before the procedure. We added this information in the Discussion section (Line 274-281).

Round 2

Reviewer 1 Report

In the manuscript, the authors present a study regarding the effects of the antithrombotic treatment on bleeding complications of EBUS-TBNA. The manuscript has been reviewed before and the authors changed the manuscript according to the previous reviewers indications. Now, the the value of the manuscript has increased. That is why, I think that this manuscript can be published in this form.

Reviewer 2 Report

Thank you for addressing my comments.